# The role of endoplasmic reticulum in *in vivo* cancer FDG kinetics

**Sara Sommariva[1], Mara Scussolini[1], Vanessa Cossu[2], Cecilia Marini[3], Gianmario Sambuceti[2,4], Giacomo Caviglia[1], Michele Piana[1,5]***

**1** Dipartimento di Matematica, Università di Genova, Genova, Italy, **2** Dipartimento di Medicina Nucleare, Policlinico San Martino IRCCS, Genova, Italy, **3** CNR - IBFM, Genova, Italy, **4** Dipartimento di Scienze della Salute, Università di Genova, Genova, Italy, **5** CNR - SPIN, Genova, Italy

* piana@dima.unige.it

**Data Availability Statement:** All relevant data and codes are deposited on GitHub at https://github.com/theMIDAgroup/BCM_CompartmentalAnalysis.git.

## Abstract

A recent result obtained by means of an *in vitro* experiment with cancer cultured cells has configured the endoplasmic reticulum as the preferential site for the accumulation of 2-deoxy-2-[$^{18}$F]fluoro-D-glucose (FDG). Such a result is coherent with cell biochemistry and is made more significant by the fact that the reticular accumulation rate of FDG is dependent upon extracellular glucose availability. The objective of the present paper is to confirm *in vivo* the result obtained *in vitro* concerning the crucial role played by the endoplasmic reticulum in FDG cancer metabolism. This study utilizes data acquired by means of a Positron Emission Tomography scanner for small animals in the case of CT26 models of cancer tissues. The recorded concentration images are interpreted within the framework of a three-compartment model for FDG kinetics, which explicitly assumes that the endoplasmic reticulum is the dephosphorylation site for FDG in cancer cells. The numerical reduction of the compartmental model is performed by means of a regularized Gauss-Newton algorithm for numerical optimization. This analysis shows that the proposed three-compartment model equals the performance of a standard Sokoloff's two-compartment system in fitting the data. However, it provides estimates of some of the parameters, such as the phosphorylation rate of FDG, more consistent with prior biochemical information. These results are made more solid from a computational viewpoint by proving the identifiability and by performing a sensitivity analysis of the proposed compartment model.

## Introduction

2-deoxy-2-[$^{18}$F]fluoro-D-glucose (FDG) is widely used as a glucose analogue radioactive tracer to evaluate glucose metabolism in living organisms. As glucose, FDG is first transported into cells, where it is phosphorylated to FDG-6-phosphate (FDG6P) by hexokinase (HK) and then accumulates intracellularly. The measured amount of emitted radiation is considered an accurate marker of overall glucose uptake by cells and tissues [1]. In addition, FDG consumption by cancer cells is increased by the Warburg effect for glucose [2]; consequently, FDG may be employed in cancer detection and staging, and to determine the effectiveness of medical treatments [3].

**Funding:** SS acknowledges the financial support of COENzYME: Chemotherapy effect On cell ENergY Metabolism and Endoplasmic reticulum redox control, granted by the Associazione Italiana Ricerca sul Cancro (AIRC). GS is the Principal Investigator for this grant.

**Competing interests:** The authors have declared that no competing interests exist.

The clinical role of FDG has been formalized in a seminal paper [4]. This paper utilizes a two-compartment model for tracer kinetics to point out that FDG competes with glucose for transmembrane transport and phosphorylation; also, it shows that the radioactivity trapped inside the cell cannot be lost for a time interval comparable to the duration of the experiment. Sokoloff's model relies on the assumption that, in most cancer lesions, FDG6P dephosphorylation occurs very slowly and therefore can be neglected in the first hour after injection. However, [5] reported that FDG6P is a substrate for G6Pase, so that it can be dephosphorylated, and that neglecting FDG6P dephosphorylation causes a systematic underestimation of the glucose consumption rate. Furthermore, recent studies have shown that G6Pase is located in the lumen of the endoplasmic reticulum (ER) [6–9]. Due to the action of the transmembrane protein glucose-6-phosphate transporter (G6PT), FDG6P enters the ER where its hydrolysis results in the creation of a phosphate group and a free molecule of FDG that is released in the cytosol [10]. The interpretation of ER as a distinct functional compartment is further supported by biochemical, pharmacological, clinical, and genetic data [11, 12].

These results have been inspirational for a recent study [13] concerning the characterization of FDG kinetics in cultured cancer cells, which shows that ER is the preferential site of FDG accumulation and that, even more importantly, the FDG reticular accumulation rate is dependent upon extracellular glucose availability. This investigation relies on two methodological tools, one experimental and one computational. In fact, on the one hand, FDG kinetics in cells cultured over a Petri dish is evaluated using the dedicated Ligand Tracer device [14], which is able to count electron/positron events without contaminating the counting rate of the cultured cells. On the other hand, the data analysis is performed within the framework of a novel compartmental model, which extends the traditional Sokoloff two-compartment analysis [4] by constraining G6Pase sequestration within the ER lumen.

However, this methodological approach leaves two significant issues open that should be addressed to provide further significance to this biochemical finding. The first issue requires the *in vivo* confirmation of the crucial role that the FDG—ER connection plays in the *in vitro* metabolism of cancer cells. The second issue is concerned with the mathematical identifiability of the three-compartment model utilized to prove this connection, which must be discussed in order to make the model sound and not ambiguous from the data analysis perspective.

The main objective of the present paper is to discuss the reliability of the proposed three-compartment model for the analysis of FDG kinetics in tissues *in vivo*. To this end we consider FDG Positron Emission Tomography (FDG-PET) data of murine models inoculated with specific cancer cells. Precisely, we have processed six datasets provided by a PET scanner for small animals in the case of six murine models of CT26 colon cancer. The compartment model used for this analysis is the analog of the one utilized in [13] for describing the FDG kinetics in the case of *in vitro* cultured data, and is designed according to the biochemically-driven assumption that most FDG is dephosphorylated in ER. As a validation, the results provided by the proposed approach are compared with those obtained through a standard Sokoloff's two-compartment model, and with the results from the *in vitro* experiment where direct verification is possible. Further, the reliability of the results is corroborated by identifiability and sensitivity considerations based on a formal and numerical analysis of the compartmental equations.

## Materials and methods

### The compartmental model

The role of ER in FDG kinetics was extended to tissues by means of the biochemically driven three-compartment model (hereon BCM) illustrated in Fig 1 (a) where $C_i$ denotes the input concentration in the arterial blood; $C_f$ is the concentration of free (not phosphorylated) FDG

(a)                                                    (b)

**Fig 1. Compartmental models used for data analysis.** (a) Biochemically-driven three-compartment model (BCM). (b) Two-compartment Sokoloff's model (SCM). $C_i$ is the concentration of the tracer in input, $C_f$ is the concentration of the free, not phosphorylated FDG, $C_p$ is the concentration of the phosphorylated FDG in the cytosol, $C_r$ is the concentration of the phosphorylated FDG in the endoplasmic reticulum.

in the cytosol; $C_p$ is the concentration of phosphorylated FDG in the cytosol; $C_r$ is the concentration of phosphorylated FDG in the ER (all concentrations are measured in kBq/ml). The rate constants $k_i$ (1/min), with $i \in \{1, 2, 3, 5, 6\}$, describe the first order process of tracer transfer between compartments: $k_1$ and $k_2$ are the rate constants for transport of FDG from blood to tissue and back from tissue to blood (by GLUTs), respectively; $k_3$ is the phosphorylation rate of FDG (by HK); $k_5$ is the input rate of FDG6P into ER (by G6PT); $k_6$ refers to the dephosphorylation rate of FDG6P to FDG (by G6Pase). Since dephosphorylation occurs only inside ER, a parameter $k_4$, corresponding to an arrow from the phosphorylated compartment to free compartment in the cytosol was not considered. Further, in principle ER may interchange free FDG molecules with the cytosol, so that free tracer can be found in ER; however, here we assumed that the free tracer in ER reaches equilibrium almost instantaneously and represents a small fraction of the tracer contained in ER, so that the input of free tracer from cytosol and the content of free tracer in ER were discarded. In Fig 1, BCM is compared to the traditional two-compartment Sokoloff model (hereon SCM); specifically, SCM is showed in panel (b), where the meaning of $C_i$, $C_f$, $C_p$ and of $k_1$, $k_2$, $k_3$, $k_4$ is analogous as in panel (a).

We assumed that standard conditions for applicability of compartmental models are satisfied. In particular, the distribution of tracer in each compartment is spatially homogeneous, and tracer exchanged between compartments is instantaneously mixed [1, 12]. We also applied appropriate corrections for the physical decay of radioactivity. Then, the time-dependent functions $C_f$, $C_p$, and $C_r$, which are the state variables of the compartmental system, the tracer concentration of the input compartment $C_i$, which is considered as the given input function (IF) for the compartmental model, and the rate constants $k_1$, $k_2$, $k_3$, $k_5$, $k_6$ are related by the system of ordinary differential equations (ODEs)

$$\frac{d}{dt}\mathbf{C}(t) = \mathbf{M}\mathbf{C}(t) + k_1 C_i(t)\mathbf{e}, \qquad \mathbf{C}(0) = \mathbf{0} , \tag{1}$$

where

$$\mathbf{M} = \begin{pmatrix} -(k_2 + k_3) & 0 & k_6 \\ k_3 & -k_5 & 0 \\ 0 & k_5 & -k_6 \end{pmatrix} , \quad \mathbf{C} = \begin{pmatrix} C_f \\ C_p \\ C_r \end{pmatrix} , \quad \mathbf{e} = \begin{pmatrix} 1 \\ 0 \\ 0 \end{pmatrix} \tag{2}$$

and with the time variable $t \in \mathbb{R}_{>0}$. The initial condition $\mathbf{C}(0) = \mathbf{0}$ means that no tracer amount is in the system at the beginning of the experiment. The analytic solution of the Cauchy problem (1) was obtained by observing that $\mathbf{C}(t) = \mathbf{0}$ is the unique solution of the homogenous

Cauchy problem corresponding to (1) and by applying the variation of constants method. In a general equation of the form

$$y' = -a(t)y(t) + f(t) \tag{3}$$

this approach looks for solutions of the form [15]

$$y(t) = c \exp(A(t)) , \tag{4}$$

where $A(t)$ is the antiderivative of $a(t)$. It follows that the solution of the inhomogeneous problem (1) is [16]

$$\mathbf{C}(t; \mathbf{k}, C_i) = k_1 \int_0^t \exp(\mathbf{M} \cdot (t - \tau)) C_i(\tau) \mathbf{e} \, d\tau , \tag{5}$$

with $\mathbf{k} = (k_1, k_2, k_3, k_5, k_6)^T \in \mathbb{R}^5_{>0}$, and upper $T$ denoting vector transposition.

We considered *in vivo* experiments consisting of sequences of PET images acquired at different time intervals. In each PET frame, two regions of interest (ROIs) were drawn, one highlighting the tumor and the other one identifying the left cardiac ventricle. The volume $V_{tot}$ of the tumor ROI was partitioned as

$$V_{tot} = V_{blood} + V_{int} + V_{cyt} + V_{er} , \tag{6}$$

where $V_{blood}$ and $V_{int}$ denote the volume occupied by blood and interstitial fluid, respectively; $V_{cyt}$ and $V_{er}$ denote the total volumes of cytosol and ER in tissue cells. The total activity in $V_{tot}$ was defined as $V_{tot}\mathcal{C}_T$, where $\mathcal{C}_T$ is the tracer concentration in $V_{tot}$. Therefore, the total activity is related to the state variables and IF by

$$V_{tot}\mathcal{C}_T = V_{blood}C_i + V_{int}C_f + V_{cyt}C_f + V_{cyt}C_p + V_{er}C_r , \tag{7}$$

or, equivalently,

$$\mathcal{C}_T = \frac{V_{blood}}{V_{tot}}C_i + \frac{V_{int} + V_{cyt}}{V_{tot}}C_f + \frac{V_{cyt}}{V_{tot}}C_p + \frac{V_{er}}{V_{tot}}C_r . \tag{8}$$

We assumed that the concentration of not phosphorylated FDG ($C_f$) is the same in the interstitial fluid and in the cytosol, while the volumes of these two sites ($V_{int}$ and $V_{cyt}$) may be different. By defining the volume fractions of blood and interstitial fluid as

$$V_b = \frac{V_{blood}}{V_{tot}} , \qquad V_i = \frac{V_{int}}{V_{tot}} , \tag{9}$$

it was straightforward to obtain

$$\frac{V_{er}}{V_{tot}} = v_r(1 - V_b - V_i) , \qquad \frac{V_{cyt}}{V_{tot}} = (1 - v_r)(1 - V_b - V_i) , \tag{10}$$

where

$$v_r = \frac{V_{er}}{V_{cyt} + V_{er}} \tag{11}$$

is independent of the number of cells. Replacement of (9) and (10) into (8) yielded the required result

$$\mathcal{C}_T = V_b C_i + \alpha_1 C_f + \alpha_2 C_p + \alpha_3 C_r , \tag{12}$$

where the positive non-dimensional constants $\alpha_1$, $\alpha_2$, and $\alpha_3$ are defined as

$$\alpha_1 = V_i + (1 - \nu_r)(1 - V_b - V_i) ,\tag{13}$$

$$\alpha_2 = (1 - \nu_r)(1 - V_b - V_i) ,\tag{14}$$

$$\alpha_3 = \nu_r(1 - V_b - V_i) .\tag{15}$$

In compact form, Eq (12) becomes

$$\mathcal{C}_T(t) = V_b C_i(t) + \boldsymbol{\alpha} C(t; \boldsymbol{k}, C_i) , \qquad \boldsymbol{\alpha} = (\alpha_1, \ \alpha_2, \ \alpha_3) ,\tag{16}$$

which describes the relationship between the acquired data and the BCM.

In the experimental applications considered in this work we set $V_b = 0.15$ according to [17] and $V_i = 0.3$ according to [18]. The volume fraction occupied by ER with respect to cytosol was computed as

$$\nu_r = \frac{V_{er}/V_{cyt}}{1 + V_{er}/V_{cyt}} \simeq 0.14 ,\tag{17}$$

where the numerical value was defined by setting the relative size of ER with respect to cytosol, $V_{er}/V_{cyt}$, equal to 0.17 as in [19, 20]. As shown in S1 Appendix, a variation up to ±50% of the chosen value of $V_{er}/V_{cyt}$ does not significantly impact the results of our analysis.

Throughout the paper the results obtained with the proposed BCM shall be compared with those from the SCM depicted in Fig 1(b). In this case the connection between the data and the SCM is expressed as

$$\mathcal{C}_T = V_b C_i + \beta_1 C_f + \beta_2 C_p\tag{18}$$

where

$$\beta_1 = \frac{V_{int} + V_{cyt}}{V_{tot}} = 1 - V_b ,\tag{19}$$

$$\beta_2 = \frac{V_{cyt}}{V_{tot}} = 1 - V_b - V_i ,\tag{20}$$

and the concentrations $C_f$ and $C_p$ are estimated by solving the ODEs system

$$\begin{pmatrix} \dfrac{d}{dt} C_f(t) \\[2ex] \dfrac{d}{dt} C_p(t) \end{pmatrix} = \begin{pmatrix} -(k_2 + k_3) & k_4 \\[1ex] k_3 & -k_4 \end{pmatrix} \begin{pmatrix} C_f(t) \\[1ex] C_p(t) \end{pmatrix} + \begin{pmatrix} k_1 C_i(t) \\[1ex] 0 \end{pmatrix},\tag{21}$$

with initial value $C_f(0) = 0$ and $C_p(0) = 0$.

## Animal models and data acquisition

We analyzed a group of six mice, denoted as m$i$, with $i = 1, \ldots, 6$, whose basic characteristics are reported in Table 1 and include: cell-line type, sex, weight, glycemia, the peak value of the arterial IF $\hat{C}_i$, reached in the first few minutes of the PET acquisition, and the tracer concentration in the ROI tissue at the end time $\mathcal{C}_T(t_f)$, where $t_f$ is the time of the last PET acquisition. All animal experiments were reviewed and approved by the Licensing and Ethical Committee of

**Table 1. Experimental values for the FDG-PET measurements.**

|  | Cell type | Sex | Weight [g] | Glycemia [mg/dL] | $\hat{C}_i$ [kBq/mL] | $\mathcal{C}_T(t_f)$ [kBq/mL] |
|---|---|---|---|---|---|---|
| m1 | CT26 | F | 18.7 | 112 | $1.52 \cdot 10^3$ | $3.18 \cdot 10^2$ |
| m2 | CT26 | F | 17.2 | 84 | $1.91 \cdot 10^3$ | $2.73 \cdot 10^2$ |
| m3 | CT26 | F | 19.9 | 162 | $9.58 \cdot 10^2$ | $1.62 \cdot 10^2$ |
| m4 | CT26 | F | 16.8 | 81 | $1.54 \cdot 10^3$ | $3.53 \cdot 10^2$ |
| m5 | CT26 | F | 17.3 | 69 | $6.69 \cdot 10^2$ | $3.44 \cdot 10^2$ |
| m6 | CT26 | F | 15.9 | 53 | $1.34 \cdot 10^3$ | $3.01 \cdot 10^2$ |

For each mouse the table shows: type of cell line, sex, weight (g), glycemia (mg/dL), maximum value of the IF denoted as $\hat{C}_i$ (kBq/mL), and final-time ($t_f$) total concentration of the cancer tissue $\mathcal{C}_T(t_f)$ (kBq/mL).

the IRCCS San Martino IST, Genova, Italy, and by the Italian Ministero della Salute. Experiments were conducted under the Guide for the Care and Use of Laboratory Animals (Italian 26/2014 and EU 2010/63/UE directives) [21], were reviewed and approved by the Licensing and Animal Welfare Body of Ospedale Policlinico San Martino of Genoa and by the Italian Ministry of Health (Ministry authorization No. 832/2016/PR), and were performed in compliance with the "ARRIVE guidelines (Animal Research: Reporting in Vivo Experiments)". The study protocol included 6-week-old male BALB/c mice (Charles River, Italy) that were fed with standard chow. Starting six hours before PET imaging animals were kept under fasting condition, with free access to water. Before imaging, mice were anesthetized with intraperitoneal administration of ketamine 100 mg/kg (Imalgene, Milan, Italy) and xylazine 10 mg/kg (Bio98, Italy). After image acquisition, mice were euthanized by cervical dislocation.

The experimental ROI concentration $\mathcal{C}_T$ of the tumor for one of the mice (specifically, mouse m1, Fig 2(a)) is shown in Fig 2(b); the related canonical arterial input function $C_i$ is shown in Fig 2(c).

In all animals we have injected 3700 KBq of FDG into the tail vein after a fasting period of six hours during a dynamic list mode acquisition ($10 \times 15s + 1 \times 122s + 4 \times 30s + 5 \times 60s + 2 \times 150s + 5 \times 300s$).

## Image analysis

The images have been reconstructed by using a standard Ordered Subset Expectation Maximization (OSEM) iterative algorithm [22] while Region of Interests (ROIs) have been drawn over the tumor tissue in order to measure its activity. ROIs have been also drawn over the left ventricle in order to compute the IF. Since the determination of IF is a particularly challenging task in the case of mice, we have first viewed the tracer first pass in cine mode; then, in a frame where the left ventricle was particularly visible, we have drawn a ROI in the aortic arc and maintained it for all time points.

FDG-PET data of CT26 cancer tissues have been processed by the application of both BCM and SCM. The numerical reduction of both models has been performed by means of a regularized Newton-type method [23, 24], already validated and applied successfully in other compartmental problems, e.g. in the modeling of complex physiologies [25, 26], in parametric imaging [27], and in reference tissue approaches [28]. The algorithm is denoted as reg-GN in the following. In order to estimate the uncertainty on the reconstructed kinetic parameters, we have computed the corresponding mean values and standard deviations over fifty runs of the reg-GN code, with fifty different initial values of the rate constants randomly drawn as described in the subsection *Sensitivity analysis*. The regularization parameter was determined

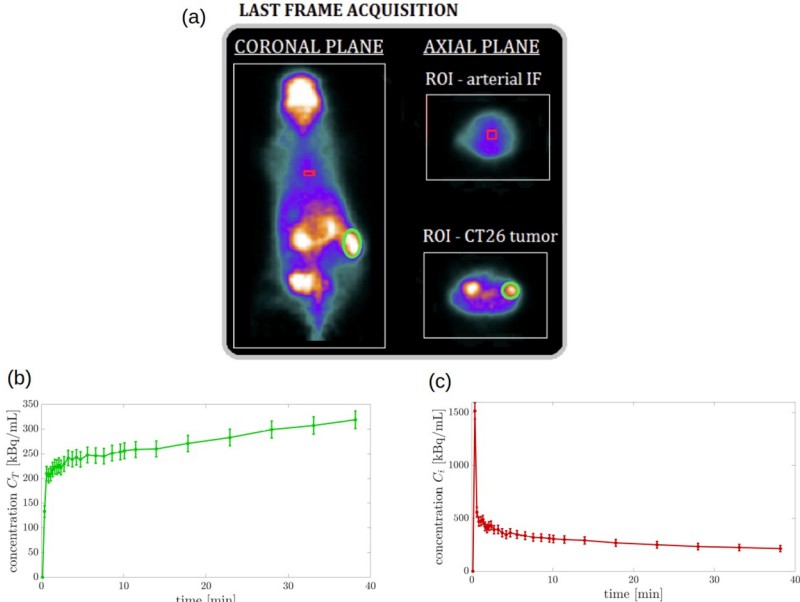

**Fig 2. FDG-PET experimental data acquired for mouse m1.** (a) Last frame of the FDG-PET acquisition of the murine model m1 with ROIs around the CT26 tumor (green color) and the aortic arc (red color). (b) The time-dependent ROI concentration curve of the CT26 tumor $\mathcal{C}_T$ and its standard deviation, related to experiment m1. (c) The time-dependent concentration curve of the arterial input function $C_i$ and its standard deviation, related to experiment m1.

at each iteration via Generalized Cross Validation method [29], with a confidence interval ranging between $10^4$ and $10^6$. The threshold for the stopping criterion of the iterative algorithm was chosen of the order of $10^{-1}$, for both models.

## Results

### Identifiability issues

Identifiability of linear compartmental models has been widely analyzed and there are a lot of results already available in the literature, some of which are collected in [30]. However, those results do not account for the fact that in experiments relying on PET images the only available measurements correspond to the tracer concentration in the left ventricle, and the overall tracer concentration in the tumor, denoted respectively with $C_i$ and $\mathcal{C}_T$ in Eq (12). Nevertheless, the standard techniques illustrated in [27, 31] and relying on the use of the Laplace transform straightforwardly inspired the proof of identifiability of BCM exhibited in S2 Appendix and summarized below.

Let us suppose that two sets $\boldsymbol{k}$ and $\boldsymbol{k}'$ lead to the same measurement $\mathcal{C}_T(t)$. This implies that

$$V_b C_i(t) + \boldsymbol{\alpha} \boldsymbol{C}(t; \boldsymbol{k}, C_i) = V_b C_i(t) + \boldsymbol{\alpha} \boldsymbol{C}(t; \boldsymbol{k}', C_i) \ . \tag{22}$$

By applying the Laplace transform to both sides of (22) and by employing (1) we obtained the necessary condition

$$k_1 \frac{Q_k(s)}{D_k(s)} = k_1' \frac{Q_{k'}(s)}{D_{k'}(s)} \ , \tag{23}$$

for all $s > 0$, where

$$Q_k(s) = \alpha_1 s^2 + [\alpha_1(k_5 + k_6) + \alpha_2 k_3]s + \alpha_1 k_5 k_6 + \alpha_2 k_3 k_6 + \alpha_3 k_3 k_5 , \tag{24}$$

$$D_k(s) = s^3 + (k_2 + k_3 + k_5 + k_6)s^2 + [(k_2 + k_3)(k_5 + k_6) + k_5 k_6]s + k_2 k_5 k_6 , \tag{25}$$

and $Q_{k'}(s)$ and $D_{k'}(s)$ are given by the same formulas but with $k'$ instead of $k$. As shown in S2 Appendix, for all $k$ up to a set of null measure in the space of parameters, the condition (23) implies $k' = k$. Therefore, according to the definition in [32], the BCM is structurally globally identifiable.

## Sensitivity analysis

The formal result concerning identifiability does not exclude the possibility of numerical non-uniqueness, which in turn would imply unreliability of the compartmental analysis. Fig 3, panels from (a) to (e), illustrates the behaviour of the total concentration $\mathcal{C}_T(t)$ computed by solving Eqs (5) and (16) for many different values of the kinetic parameters. To choose reasonable ranges for the parameters, we started from the values provided by the numerical solution of the inverse problem for a representative mouse (mouse m1 in Table 2). Namely we defined the reference values $\hat{k}_1 = 0.32, \hat{k}_2 = 0.37, \hat{k}_3 = 0.45, \hat{k}_5 = 0.51, \hat{k}_6 = 0.03$. In the different panels we plotted the curves obtained by varying one parameter at the time so to assume 20 different values logarithmically spaced in the interval $\hat{k}_j \cdot [10^{-1}, 10^2]$. In all the simulations we used the IF measured for mouse m1.

These plots show that $k_5$ and $k_6$ are the critical parameters in the set. Indeed, the different values of $k_6$ produced curves of the total concentration that were close to each other. Instead, values of $k_5$ lower than $\hat{k}_5$ generated well distinguishable curves, suggesting a good sensitivity of $\mathcal{C}_T$ to small values of $k_5$. However such sensitivity rapidly decreased when $k_5$ approached and exceeded the reference value.

We also performed a more rigorous sensitivity analysis [33] by computing the relative local sensitivity $S_T$ of $\mathcal{C}_T$ versus the five kinetic parameters in BCM. Specifically, for each parameter $k_j$ we considered many different values obtained by scaling the reference value $\hat{k}_j$ by a factor in the range $[10^{-3}, 10^3]$. For all these values we computed the curve

$$S_T^j(t) := |k_j \frac{d\mathcal{C}_T(t)/dk_j}{\mathcal{C}_T(t)}| \tag{26}$$

with $\mathcal{C}_T(t)$ determined by solving the problem described by Eqs (5) and (16). In panel (f) of Fig 3 we plotted the corresponding asymptotic value for $t = 40$ min.

The plot in panel (f) of Fig 3 confirms that $k_5$ and $k_6$ are characterized by the lowest sensitivity. Specifically, on the one hand, the sensitivity of $k_1$, $k_2$ and $k_3$ is high for values of the scaling factor close to one, i.e. for values of the parameters close to the ones provided by the numerical solution of the inverse problem. On the other hand, the sensitivity of $k_6$ is very small for values close to the solution of the inverse problem, and the one of $k_5$ rapidly decreases for values bigger than this solution.

To account for these results we introduced a constraint on the values used to initialize $k_5$ and $k_6$ within the reg-GN algorithm. In detail, we initialized $k_5$ according to the following simple heuristic procedure. First we processed the same experimental data by using the standard SCM, which provides a first estimate of the four kinetic parameters $k_1, k_2, k_3, k_4$; then, in BCM, we set the initial value of $k_5$ equal to a random positive number lower than

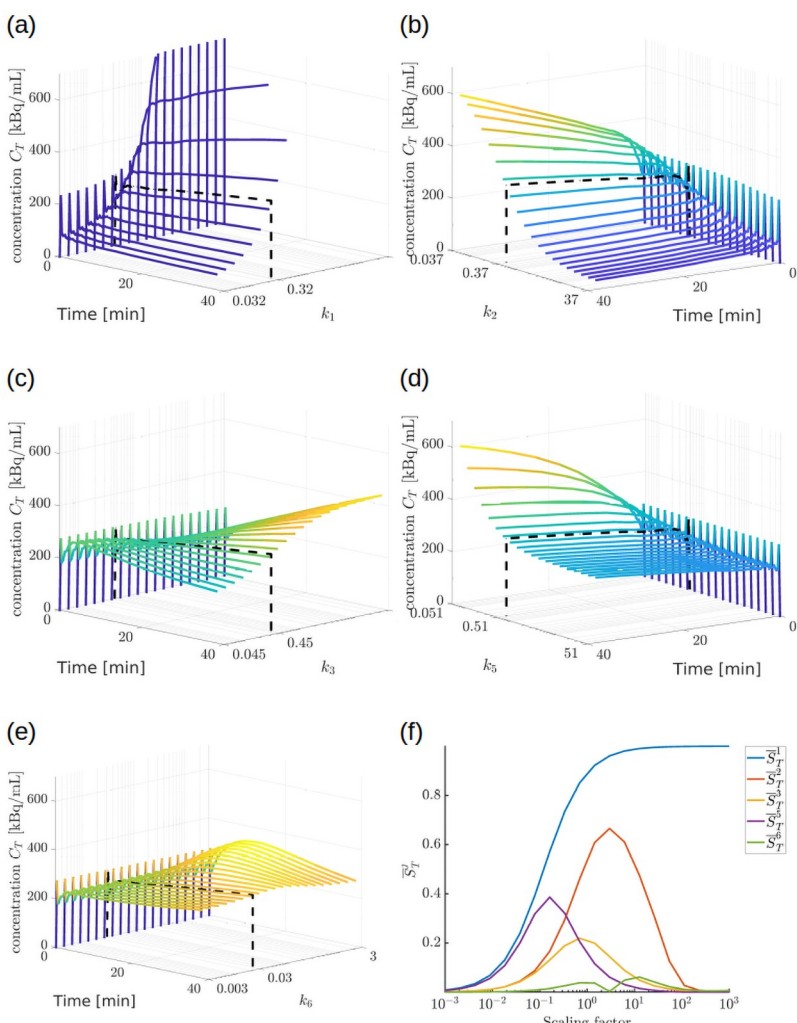

**Fig 3. Sensitivity analysis for the five kinetic parameters utilized in BCM.** (a)-(e) Behaviour of the total concentration $\mathcal{C}_T(t)$ computed using Eqs (5) and (16) for several values of the kinetic parameters. Each panel refers to a different parameter $k_j$. Specifically, the black dotted curve has been computed by using the values of the parameters provided by solving the compartmental inverse problem for mouse m1. The other curves have been obtained by modifying only the parameter $k_j$. Notice that the axis in the different panels present different orientations so as to improve the visibility of the plotted curves. (f) Asymptotic value $\bar{S}_T^j$ of $S_T^j$ in Eq (26) for $t = 40$ min. Each curve depicts the value of $\bar{S}_T^j$ for a different kinetic parameter $k_j$ as function of its normalized value $k_j/\hat{k}_j$ (scaling factor).

$\sqrt{\left(k_1\right)^2 + \left(k_2\right)^2 + \left(k_3\right)^2 + \left(k_4\right)^2}$, where the values of these four parameters were estimated by SCM. This initialization condition implements an energetic constraint on $k_5$, which prevents it from reaching ranges of values where the sensitivity of $\mathcal{C}_T$ becomes too small. Instead, the initial value of $k_6$ was set equal to 0 in order to promote small estimates of such a parameter. This choice was supported by results shown in previous works where the dephosphorylation rate of FDG6P was assumed to be zero [4, 34] or estimated of the order up to $10^{-2}$ (1/min) [35, 36].

The initial values of $k_1$, $k_2$, $k_3$ of BCM and of the four parameters of SCM, were randomly drawn in (0, 1).

**Table 2. Reconstructed kinetic parameters (1/min) of BCM.**

|  | $k_1$ | $k_2$ | $k_3$ | $k_5$ | $k_6$ |
|---|---|---|---|---|---|
| m1 | 0.32 ± 0.03 | 0.37 ± 0.15 | 0.45 ± 0.19 | 0.51 ± 0.28 | 0.03 ± 0.02 |
| m2 | 0.47 ± 0.04 | 0.67 ± 0.14 | 0.54 ± 0.16 | 0.59 ± 0.26 | 0.03 ± 0.04 |
| m3 | 0.17 ± 0.03 | 0.34 ± 0.17 | 0.58 ± 0.21 | 0.56 ± 0.25 | 0.03 ± 0.02 |
| m4 | 0.25 ± 0.03 | 0.22 ± 0.12 | 0.64 ± 0.21 | 0.58 ± 0.23 | 0.08 ± 0.02 |
| m5 | 0.30 ± 0.04 | 0.33 ± 0.19 | 0.85 ± 0.31 | 0.61 ± 0.27 | 0.09 ± 0.05 |
| m6 | 0.31 ± 0.03 | 0.36 ± 0.12 | 0.61 ± 0.21 | 0.53 ± 0.27 | 0.09 ± 0.03 |
| mean | 0.30 | 0.38 | 0.61 | 0.56 | 0.06 |
| std | 0.09 | 0.15 | 0.13 | 0.04 | 0.03 |

Values of the kinetic parameters for the CT26 tumor tissue estimated by applying BCM on a FDG–PET experimental group of six mice. The first six rows show mean and standard deviation over 50 runs of the reg-GN algorithm. The last two lines report mean and standard deviation of each kinetic parameter computed over the mean estimates of the six murine experiments.

## Tracer kinetics

The application of reg-GN to estimate the vector **k** of tracer coefficients leaded to the results reported in Table 2 for BCM and Table 3 for SCM.

After reconstructing the parameter vector *k*, we computed the compartment concentrations by solving the Cauchy problem associated to BCM, Eq (1), and SCM, Eq (21). Then we reconstructed $\mathcal{C}_T$ by using Eqs (12) and (18), respectively. Fig 4 shows the concentration curves for mouse m1, which has been chosen as representative of all the FDG-PET CT26-tissue experiments. In particular, panels (a) and (b) demonstrate that the reconstructed concentration $\mathcal{C}_T$ fit the experimental data for both models. Specifically, over 50 different estimates provided by 50 repetitions of the reg-GN, the relative error ($\ell_2$-norm over time) between the acquired and the reconstructed $\mathcal{C}_T$ was always below 0.13 for BCM and 0.17 for SCM.

## Discussion

### Robustness of the numerical method

Various results shown in the previous section corroborate the reliability of the proposed reg-GN method for model reduction of BCM and SCM.

**Table 3. Reconstructed kinetic parameters (1/min) of SCM.**

|  | $k_1$ | $k_2$ | $k_3$ | $k_4$ |
|---|---|---|---|---|
| m1 | 0.32 ± 0.02 | 0.62 ± 0.09 | 0.13 ± 0.09 | 0.03 ± 0.04 |
| m2 | 0.43 ± 0.02 | 0.84 ± 0.06 | 0.11 ± 0.01 | 0.03 ± 0.01 |
| m3 | 0.16 ± 0.03 | 0.60 ± 0.19 | 0.14 ± 0.06 | 0.03 ± 0.02 |
| m4 | 0.23 ± 0.02 | 0.57 ± 0.11 | 0.26 ± 0.14 | 0.04 ± 0.03 |
| m5 | 0.28 ± 0.02 | 0.67 ± 0.12 | 0.23 ± 0.05 | 0.03 ± 0.01 |
| m6 | 0.30 ± 0.02 | 0.64 ± 0.09 | 0.23 ± 0.05 | 0.05 ± 0.01 |
| mean | 0.29 | 0.66 | 0.18 | 0.03 |
| std | 0.09 | 0.09 | 0.06 | 0.01 |

Values of the kinetic parameters for the CT26 tumor tissue estimated by applying SCM on a FDG–PET experimental group of six mice. The first six rows show mean and standard deviation over 50 runs of the reg-GN algorithm. The last two lines report mean and standard deviation of each kinetic parameter computed over the mean estimates of the six murine experiments.

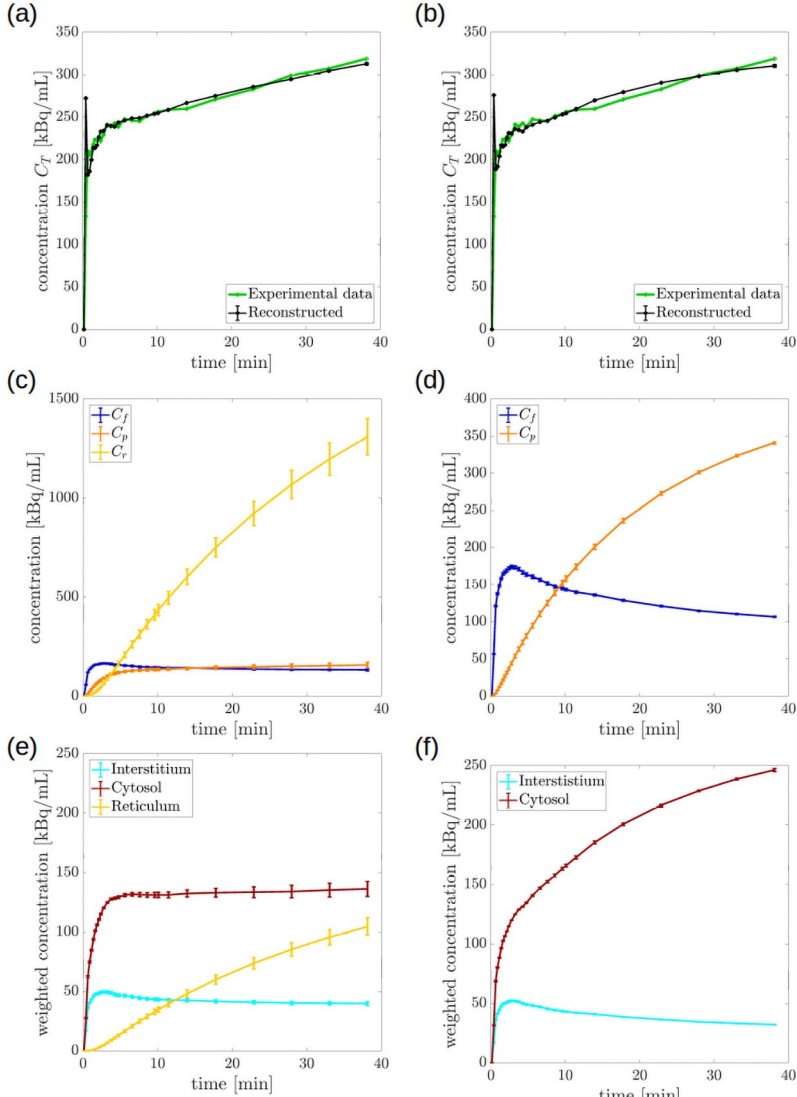

**Fig 4. Model-predicted time curves of the ROI concentration and of the compartment activity for mouse m1.** (a, b) Experimental and reconstructed curve of the concentration $\mathcal{C}_T$ in BCM and SCM, respectively; (c) $C_f$, $C_p$ and $C_r$ of BCM; (d) $C_f$, and $C_p$ of SCM; (e) contribution to the total concentration due to tracer in the interstitial fluid, $\frac{V_{int}}{V_{tot}}C_f$, in the cytosol, $\frac{V_{cyt}}{V_{tot}}\left(C_f+C_p\right)$, and in the ER, $\frac{V_{er}}{V_{tot}}C_r$ of BCM; (f) contribution to the total concentration due to tracer in the interstitial fluid, $\frac{V_{int}}{V_{tot}}C_f$, and in the cytosol, $\frac{V_{cyt}}{V_{tot}}\left(C_f+C_p\right)$, of SCM. All the panels show average and standard error of the mean of the curves predicted across 50 repetitions of the reg-GN algorithm.

First of all, as shown in Fig 4(a) and 4(b), the values of the rate constants estimated through the inversion procedure produced a total concentration that approximated well the measured concentration $\mathcal{C}_T$.

Furthermore, the reconstructed values of the kinetic parameters in Tables 2 and 3 are rather stable with respect to the considered murine models, which seems to imply that these numbers describe characteristic kinetic properties of FDG inside the CT26 tissue, independently of the specific murine experiment. We also tested the reliability of the reconstructed values in BCM, taking into account the fact that the volume ratio $V_{er}/V_{cyt}$ is not known exactly. Specifically,

the inversion procedure was applied under the assumption that the ratio was misestimated up to ±50% of the chosen reference value. The results in S1 Appendix show that the reconstructed values of the rate constants were rather stable, indicating that the results obtained by application of BCM are only slightly sensitive to reasonable changes of the value of the parameter $V_{er}/V_{cyt}$ provided to the reg-GN algorithm.

As a final remark, we observe that the results of the analysis based on SCM are coherent with the ones obtained in the literature in the case of other kinds of carcinoma, for both mice and humans [37, 38]. See the next section for further details.

## Kinetic parameters and comparison with previous works

Tables 2 and 3 summarize the estimated values of the kinetic parameters, thus allowing a quantitative comparison between the two compartmental models, BCM and SCM, while explaining the same experimental data.

More in details, both models provided a similar estimate for $k_1$, of about 0.30 (1/min). This correspondence agrees with the fact that $k_1$ regulates the input flow from the blood to the tissue and the cells, which should not depend on the employed model. In fact, a similar result was also found in a previous experiment *in vitro* based on cancer cultured cell [13].

On the contrary, the two compartment models provided rather different values for $k_2$. This fact is due to the different coefficients ruling the contribution of $C_f$ to the total concentration $\mathcal{C}_T$, namely $\alpha_1 = 0.77$ for BCM and $\beta_1 = 0.85$ for SCM, and reflects the dissimilar spatial distribution of free and phosphorylated tracer assumed in the two models. Clearly this effect was not visible in the experiment *in vitro* analysed in [13] where the two compartment models only focused on the cell volume, neglecting blood and interstitial volume. Indeed, in that case BCM and SCM provided almost equal estimates of $k_2$.

As far as $k_3$ is concerned, by applying SCM we estimated a value of 0.18 (1/min), while BCM provided a remarkably higher value of about 0.61 (1/min). A similar result was also obtained from the analysis *in vitro* of [13]. In that case, the results were validated by estimating $k_3$ through a procedure independent of compartmental modeling. Precisely, the phosphorylation rate of HK, with FDG as a substrate was estimated by means of the Michaelis-Menten law [39, 40]. The resulting value $k_3 = 0.90 \pm 0.13$ min$^{-1}$ was found to be significantly closer to the one obtained by BCM kinetics, in comparison with that of SCM kinetics. We conclude that the two models provide a comparable data fitting, but BCM provides a phosphorylation rate more consistent with biochemical data.

Finally, the values obtained for $k_6$ in BCM are comparable to those of $k_4$ in SCM, consistently with the fact that both parameters describe the rate of hydrolysis of FDG6P by G6Pase, a process that is expected to be only mildly dependent on the model. Moreover, the reconstructed values of $k_6$ and $k_4$ were of the order of $10^{-2}$, in agreement with previous results reporting a very slow process of hydrolysis [13, 35, 36]. We also remark that in a few previous works, based on the Sokoloff's model, $k_4$ was assumed to be negligible [4, 34, 37] especially in experiments lasting less than two hours [41]. Here we preferred to avoid such an assumption, as it was demonstrated [5] that setting $k_4 = 0$ led to an underestimate of the metabolic rate of FDG, and hence of glucose. Furthermore, Fig 4(d) shows that the contribution $k_4 C_p$ to the ODEs for tracer balance in the SCM cannot be discarded *a priori*, since it becomes comparable to other contributions when $C_p$ grows with time. All these comments also apply to $k_6$, with the additional remark that dephosphorylation of FDG6P occurs inside the lumen of ER, thus leading to deep changes in the overall kinetics of tracer in tissues, Fig 4(c).

Additionally, we observe that the results obtained with the SCM are coherent with those provided by previous studies on mice with prostate carcinoma xenograft [37], and on soft

tissue carcinomas in human patients [38]. Similar values, even though systematically lower, were obtained also by applying the traditional Sokoloff's model for the analysis of cerebral metabolism in albino rats [4], and healthy human subjects [34, 35]. The small differences in the estimated values of the kinetic parameters may be understood by observing that traditional Sokoloff models adopted by various authors may be recovered from the present SCM by the the introduction of specific constraints as $V_i = 0$, $V_b = 0$, or $k_4 = 0$.

## Time curves of the compartment concentrations

In the following we shall refer to mouse m1, which has been chosen as representative of all FDG-PET CT26-tissue experiments. Substitution of the estimated values of the rate constants into the system of ODEs (1) and (21), followed by the solution of the related direct problems, allows completing the analysis of tracer kinetics. In particular, panels (c) and (d) of Fig 4 show the reconstructed time courses of the concentrations $C_f$, $C_p$, $C_r$ of BCM, and $C_f$, $C_p$ of SCM, respectively. Please notice the different scales of the vertical axes.

We observe that both compartmental models estimated a time curve of the free tracer similar to those found, for example, in [38]. Namely, $C_f$ decreased rather slowly after an initial peak reached in the first few minutes, Fig 4(c) and 4(d). Furthermore, the reconstruction obtained with the SCM indicated that accumulation of tracer takes place in the cytosolic phosphorylated pool. On the contrary, for BCM, the concentration $C_p$ of the phosphorylated tracer in cytosol showed an almost stationary behavior, where the equilibrium point was reached in the first minutes; instead, the phosphorylated tracer accumulated in the ER.

We observe that, at each time $t$ the value of $C_r(t)$ in BCM, Fig 4(c), was almost four times those of $C_p(t)$ in SCM, Fig 4(d). This behavior follows from the fact that the ER compartment in the BCM approach was assumed to occupy a much smaller volume than the cytosolic phosphorylated compartment in SCM, thus causing a growth of the related concentration, although the total activities were comparable. To support this interpretation, in panels (e) and (f) we plotted the contributions of each compartment to the total concentration $\mathcal{C}_T$: interstitium, cytosol and ER for BCM, interstitium and cytosol for SCM. As expected on physiological grounds, the two models showed the same interstitial concentration, while the total cytosolic concentration in SCM equaled the sum of cytosolic and reticular concentrations in BCM.

The results shown in Fig 4 are consistent with those obtained with an *in vitro* analysis of cultured cancer cell [13]. Indeed, in the latter case the reconstruction procedure showed accumulation of FDG6P in ER for BCM, and accumulation in cytosol for SCM. In addition, the accumulation of radioactivity in ER was also confirmed by direct imaging, thus providing a further indication of the better reliability of BCM in the analysis of diffusion of FDG in tissues. Both pairs of panels, (c) and (d), or (e) and (f) give an almost immediate insight into the changes induced by the introduction of ER in the representation of tracer kinetics in cancer cells.

## Limitations of BCM

We presented a biochemistry-based compartmental model for tracer kinetics, aimed at providing a more realistic description of FDG metabolism by introducing the ER as the compartment where dephosphorylation of FDG6P occurs. Naturally, the introduction of a new compartment results in a more complicated mathematical scheme, with additional parameters to be estimated or imported. More in detail, we proved that the proposed model is identifiable, but a preliminary sensitivity analysis clearly showed that the sensitivity of $k_5$ and $k_6$, i.e. of the input rate of FDG6P into ER and of its rate of dephosphorylation, is low, so that the determination of these crucial kinetic parameters by the given experimental data may be affected by

numerical non-uniqueness. In the present version of the reg-GN algorithm, we coped with this issue by properly tuning the intervals where the proposed iterative approach draws the initial values of $k_5$ and $k_6$, according to biological evidences from previous works. Future effort will be devoted to develop different inversion procedures capable of automatically selecting such intervals based, for example, on a global sensitivity analysis [42].

In the present work, the new model has been applied to the analysis of PET data coming from CT26 cancer tissues. If compared to the standard Sokoloff's model, BCM did not increase the experimental fidelity with respect to these data. However, the corresponding estimated tracer kinetics showed a better consistency with respect to previous biochemical evidence. In fact, previous studies *in vitro* reported a high colocalization of the fluorescent signals emitted by the FDG analogue 2-[N-(7-nitrobenz-2-oxa-1,3-diazol-4-yl)amino]-2-deoxyglucose (2-NBDG) and by the ER probe glibenclamide, respectively, in the ER lumen [13]. This placement and its tight correlation with FDG uptake were found to be strictly linked with the activity of the reticular enzyme hexose-6P-dehydrogenase(H6PD) in cell cultures of breast and colon carcinoma [10] as well as in neurons and astrocytes [9]. Additionally the phosphorylation rate $k_3$ estimated with BCM was found to be strongly coherent with some measurement of this enzymatic activity. All these facts seem to corroborate the superiority of BCM despite the lack of an *in vivo* direct experimental verification of FDG accumulation in ER.

## Conclusions

This paper has shown that an accumulation of FDG in phosphorylated form in ER is compatible with FDG-PET data recorded from animal models of CT26. This result has been achieved by introducing a novel compartment model where the two available compartments for phosphorylated tracer, cytosolic and ER-localized, were treated on the same level, with no *a priori* constraints imposed to the model.

A similar result was already observed for cancer cell cultures *in vitro* [13]. However, the experimental setting considered here is highly different with respect to the one in [13]: (i) cell cultures and tissues have been inserted in different environments (clean incubation medium vs heterogeneous background, including blood and interstitial tissue), are constituted by different type of cancer cells (4T1 vs CT26), and occupy different total volumes; (ii) the datum of tracer consumption has been obtained through processes based on direct measurements of the emitted radiation (LT signals) and analysis of reconstructed images of radioactivity distribution (PET images); (iii) the IF of the cell system is almost constant, while the IF of the tissue system shows a sharp peak at the initial time. Despite all these discrepancies, the reconstructed FDG kinetics obtained *in vivo* showed many characteristics similar to those arising from the *in vitro* experiment. The similar performance of BCM in such dissimilar environments further suggests the reliability of the proposed model. In this regard, we also remark that the BCM model has been shown to be identifiable, while the numerical analysis is rather robust.

Throughout the paper, the performance of BCM has been compared with that of a Sokoloff type compartmental model applied on the same set of data, consisting of the input function and the total concentration in a given ROI. Both models provide a satisfactory approximation of the experimental data but BCM has been shown to be more adherent to biochemical a priori information. In details the BCM yielded tracer accumulation in ER lumen, as it is expected from *in vitro* experiments, and provided a better estimate, e.g., of the phosphorylation rate $k_3$, describing the action of the enzyme HK.

The results provided by the application of BCM need for further investigations. In fact, the basic scheme of this approach is rather flexible and may be modified to allow for consideration of peculiarities of specific organs, as done for classical models in [25, 26], may be associated

with reference tissue formulations (see [28] and references cited therein), or to pixel-wise analysis (see [27] and references cited therein).

## Supporting information

**S1 Appendix. Robustness of the proposed reg-GN algorithm to the value of the volume fraction occupied by ER with respect to cytosol.**
(PDF)

**S2 Appendix. Proof of the identifiability of BCM.**
(PDF)

## Author Contributions

**Conceptualization:** Cecilia Marini, Gianmario Sambuceti, Giacomo Caviglia, Michele Piana.

**Data curation:** Sara Sommariva, Mara Scussolini, Vanessa Cossu, Cecilia Marini, Gianmario Sambuceti.

**Formal analysis:** Sara Sommariva, Mara Scussolini, Vanessa Cossu, Giacomo Caviglia.

**Funding acquisition:** Gianmario Sambuceti.

**Methodology:** Sara Sommariva, Vanessa Cossu, Cecilia Marini, Gianmario Sambuceti, Giacomo Caviglia.

**Software:** Sara Sommariva, Mara Scussolini.

**Supervision:** Gianmario Sambuceti, Giacomo Caviglia.

**Validation:** Sara Sommariva, Mara Scussolini, Vanessa Cossu, Cecilia Marini, Gianmario Sambuceti, Michele Piana.

**Writing – original draft:** Michele Piana.

**Writing – review & editing:** Sara Sommariva, Cecilia Marini, Gianmario Sambuceti, Giacomo Caviglia, Michele Piana.

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
