## [Decision Letter · Decision Letter 0]

22 Feb 2021

PONE-D-21-01285

The role of endoplasmic reticulum in in vivo cancer FDG kinetics

PLOS ONE

Dear Dr. piana,

Thank you for submitting your manuscript to PLOS ONE. After careful consideration, we feel that it has merit but does not fully meet PLOS ONE’s publication criteria as it currently stands and needs a very substantial revision. Therefore, we invite you to submit a revised version of the manuscript that addresses the points raised during the review process.

We look forward to receiving your revised manuscript.

Kind regards,

Domokos Máthé

Academic Editor

PLOS ONE

Additional Editor Comments:

Dear Authors,

I, as Editor, very much respect the route where you embarked. As you see the reviewers were substantially criticizing your work and rightly so, especially in not seeing the results of real-world measurements and not finding well substantiated references for estimations. However I think your work is highly important for both the biochemistry, the cell biology and the broader Nuclear Medicine/Molecular Imaging community and your basic concept is worth pursuing. So I propose a very substantial major revision instead of rejecting the manuscipt. Please especially pay attention to deal with the statements regarding the BCM model vs. the SCM model and the data available with 14C-glucose, 3H-glucose from 30 years ago, or your own experimental measurements in vitro, not just estimations.

I do look forward to receiving your substantially rewritten major-revision.

Cordially,

Domokos Máthé PhD

Academic Editor

Journal Requirements:

2. Please provide the same information from your Ethics Statement in the Methods section of the manuscript.

4. We noticed you have some minor occurrence of overlapping text with the following previous publication, which needs to be addressed:

https://ieeexplore.ieee.org/document/9140512 (Introduction, paragraph 2, sentences 5-6)

In your revision ensure you cite all your sources (including your own works), and quote or rephrase any duplicated text outside the methods section. Further consideration is dependent on these concerns being addressed.

Reviewers' comments:

Reviewer's Responses to Questions

**Comments to the Author**

1. Is the manuscript technically sound, and do the data support the conclusions?

Reviewer #1: No

Reviewer #2: No

2. Has the statistical analysis been performed appropriately and rigorously? 

Reviewer #1: No

Reviewer #2: No

3. Have the authors made all data underlying the findings in their manuscript fully available?

Reviewer #1: Yes

Reviewer #2: Yes

4. Is the manuscript presented in an intelligible fashion and written in standard English?

Reviewer #1: Yes

Reviewer #2: No

5. Review Comments to the Author

Reviewer #1: The main difficulty with the problem in the article is that on a macroscopic measurement like PET for concentration, it is not possible to provide data for phosphorilated, non phosphorilated, cytosol and ER fractions. Therefore only C_T can be used.

This precludes the claim of the authors that the BCM model can be proven to be better fit than the SCM model.

When determining the volume fraction of ER to cytoplasma, v_r, in the article it must be referenced, or reasoned of why the value given v_r = 0.14. The text says estimated as 0.14, however the specific value will determine the exact C_er and C_c concentrations heavily, thus this estimate has to be very solid. Was it optimized together with the k_i parameters? Or taken as typical values from cellbiology of these cancer cell lines? Or used as a hand-tuned hyperparameter that gives the best fit?

"A straightforward computational analysis showed.....very few and totally unrealistic". You have to provide numerical probabilities for "very few", and numbers for the physiological ranges that these exceed thus becoming "unrealistic".

k5 and k6 are somewhat underdetermined parameters, and also depend on the volume fraction of ER to total cell. So these parameters having 2 undetermined values will of course give a huge plane of solutions yielding the same C_T. The initial condition and soft bound method is okay to work with, though, but see the paragraph about cellbiological constraints of my review on relevant ranges.

You also have to provide numerical values for the k6 impact and provide numerical estimate for stating it is negligible.

On a more general note, Figure 3 should be redesigned and much better reflect multiple parameter dimension sweeps. At the moment this is a very small area of the 6D parameter space that is shown. These graphs have to be convincing that the results are robust to the qualitative parameter transitions (e.g. 3(c) inversion at time = 10 min).

I would also like to see some of the constraints and results presented in this macroscopic in vivo article explained numerically from the in vitro study, where accumulation of cellular compartments is possible. Even if FDG is not available, Glucose and Gl6H typical cellphysiological numerical values should guide both the constraints of the model, and the results should be compared to. This is essential to show that the BCM model indeed captures cellphysiologically relevant fractional compartmental kinetics, rather than arbitrarily fitting a high order polynomial to a nonlinear C_T graph. In fact the more biological evidence you can marshall behind specific values of the chosen parameters and the results, the fitted and found parameters, the less is required to evidence that the method is a relevant better model than the two compartment model.

The article must contain significantly detailed description of how the measured concentration data is consolidated in space to fit the model. At the moment there is nothing about it.

Fig 4 shows a very contradictory image of the comparison of the BCM and SCM models. First the concentration graphs should include the total concentration per tissue volume, C_T, as well as the data so that it can be seen that they fit well. Second the detailed concentration graphs should also including interstitial and blood fractions individually. Third the grand total of the intracellular concentrations in these graphs are not comparable. In fact they converge to completely different cellular total concentrations differing in a factor of 3-4. The models should not be different with cell in and out parameters, nor with the interstital and blood concentrations, and therefore not with the intracellular concentrations. I recommend a total volume check as well.

The article contains several logical threads that are repeated unnecessarily in different parts of the article. On the other hand many other thoughts are missing or poorly evidenced. The article should also see a native english speaker correction.

All of the above are very serious blocking issues. The authors should provide significant and satisfactory improvement of the presentation of their material, otherwise this article cannot be passed as a valid article. Major revision here means a really really major revision.

Reviewer #2: The main problem in this article is that the measurement like PET for concentration, it is not at all possible to provide data for phosphorylated, non-phosphorylated, cytosol and ER fractions.

The authors failed to provide significant and justified presentation of their data and content!

The Figure 3 should be reformulated in much much better understanding for multiple parameter dimension. Practically this is a very little area of the 6D parameter space that has been shown

This article needed a Major revision in all respects

The article also need native English correction.

6. PLOS authors have the option to publish the peer review history of their article (what does this mean?). If published, this will include your full peer review and any attached files.

Reviewer #1: No

Reviewer #2: No

---

## [Author Response · Author response to Decision Letter 0]

6 Apr 2021

Dear Editor, 

We greatly appreciate your interest in our work and both reviewers’ effort in reading our manuscript and providing constructive suggestions. 

Following their comments we substantially revised our manuscript. In particular, we have more carefully interpreted the obtained results in light of our own in vitro experiment as well as of previous studies employing a Sokoloff’s compartment model similar to the one used here. A point-by-point answer to reviewers' comments can be found in the Response letter attached to this submission.

We also addressed the Journal's requirement. In detail:

1) We renamed the figure files to better adhere with PLOS ONE's style requirements. 

2) The ethic statement has been reported in the Method section (pag 6 lines 138-151 of the new version of the manuscript)

3) All relevant data and code are now deposited at the publicly available git-hub repository https://github.com/theMIDAgroup/BCM_CompartmentalAnalysis.git.

4) We revised the text that was overlapping with our previous publication, see page 2 lines 19-24. We point out that such a publication was already cited in the manuscript. However we now cite it also in the aforementioned part of the text. We are sorry for the inconvenience.

Thank you for considering our work, 

The authors

---

## [Decision Letter · Decision Letter 1]

17 May 2021

The role of endoplasmic reticulum in in vivo cancer FDG kinetics

PONE-D-21-01285R1

Dear Dr. piana,

We’re pleased to inform you that your manuscript has been judged scientifically suitable for publication and will be formally accepted for publication once it meets all outstanding technical requirements.

Kind regards,

Domokos Máthé

Academic Editor

PLOS ONE

Additional Editor Comments (optional):

Reviewers' comments:

Reviewer's Responses to Questions

**Comments to the Author**

1. If the authors have adequately addressed your comments raised in a previous round of review and you feel that this manuscript is now acceptable for publication, you may indicate that here to bypass the “Comments to the Author” section, enter your conflict of interest statement in the “Confidential to Editor” section, and submit your "Accept" recommendation.

Reviewer #1: All comments have been addressed

Reviewer #2: All comments have been addressed

2. Is the manuscript technically sound, and do the data support the conclusions?

Reviewer #1: Yes

Reviewer #2: Yes

3. Has the statistical analysis been performed appropriately and rigorously? 

Reviewer #1: Yes

Reviewer #2: Yes

4. Have the authors made all data underlying the findings in their manuscript fully available?

Reviewer #1: Yes

Reviewer #2: Yes

5. Is the manuscript presented in an intelligible fashion and written in standard English?

Reviewer #1: Yes

Reviewer #2: No

6. Review Comments to the Author

Reviewer #1: The authors made a substantial effort to amend the fundamental issues with presenting their material in that form.

In particular

- the biophysical justification of hyperparameter choices and parameter constraints using the literature and their previous in vitro study is now sufficient

- the parameter sweep and robustness analyses are now sufficient

- they have addressed the ambiguity of concentration volume and activity for the models, and they are now comparable

- they have addressed introduction, claims, discussion extensively to better reflect the nature of the estimation of a

directly unmeasurable compartment concentration proportion

The manuscript is a go from me. Thank you for taking due diligence and effort. This is now a sufficiently well evidenced paper.

Reviewer #2: The author has adequately addressed the comments raised by the reviewers! Now this manuscript looks fine.

7. PLOS authors have the option to publish the peer review history of their article (what does this mean?). If published, this will include your full peer review and any attached files.

Reviewer #1: No

Reviewer #2: No

---

## [Editor Report · Acceptance letter]

20 May 2021

PONE-D-21-01285R1 

The role of endoplasmic reticulum in *in vivo* cancer FDG kinetics 

Dear Dr. Piana:

I'm pleased to inform you that your manuscript has been deemed suitable for publication in PLOS ONE. Congratulations! Your manuscript is now with our production department. 

Kind regards, 

on behalf of

Dr. Domokos Máthé 

Academic Editor

PLOS ONE